# Identifying Behavior Change Techniques Used in Tobacco Cessation Interventions by Oral Health Professionals and Their Relation to Intervention Effects—A Review of the Scientific Literature

**DOI:** 10.3390/ijerph18147481

**Published:** 2021-07-13

**Authors:** Ibtisam Moafa, Ciska Hoving, Bart van den Borne, Mohammed Jafer

**Affiliations:** 1Department of Preventive Dental Sciences, Jazan University, Jazan 45142, Saudi Arabia; dr.mjafer@gmail.com; 2Department of Health Promotion, Maastricht University, 6200 MD Maastricht, The Netherlands; c.hoving@maastrichtuniversity.nl (C.H.); b.vdborne@maastrichtuniversity.nl (B.v.d.B.)

**Keywords:** taxonomy, oral health, behavior change, tobacco cessation, dental practice, interventions

## Abstract

This review aimed to identify the behavioral change techniques (BCTs) used in behavioral interventions for tobacco cessation at dental practices in relation to their effect on tobacco use. Six scientific databases were searched for behavior change interventions for tobacco cessation and were coded using the BCT taxonomy of behavioral support for smoking cessation (BCTTsm). Fifteen interventions were identified, and data related to intervention characteristics were abstracted. Sixteen BCTs were identified, mainly related to increased motivation and teaching regulatory skills. Goal setting was the most commonly used BCT. Ten out of fifteen interventions effectively impacted tobacco cessation outcomes (OR = 2 to 5.25). Effective interventions more frequently included goal setting, written materials, readiness to quit and ability assessment, tobacco-use assessment, self-efficacy boost, listing reasons for quitting, action planning and environment restructuring. Other BCTs were not clearly associated with an increased effect. Among the behavioral interventions, certain techniques were associated with successful tobacco quitting. Tobacco cessation interventions in a dental setting appear to benefit from using BCTs that increase motivation and teach regulatory skills. The identified BCTs in this review could provide a source to better inform researchers and dentists about the active ingredients in behavior change interventions for tobacco cessation in a dental setting.

## 1. Introduction

The dental setting has a unique position in tobacco cessation interventions. The repeated nature and long duration of dental appointments offer multiple opportunities for oral health professionals to educate and motivate their patients to quit tobacco use [1]. Evidence shows that adults tend to visit oral healthcare clinics more than visiting their physician [2,3]. Furthermore, dental patients perceive oral health professionals as a credible source for tobacco cessation interventions [1]. Tobacco users face many oral health issues and frequently visit dental clinics. Thus, dental professionals should know more about which behavior change techniques (BCTs) work and which ones do not. Furthermore, the deleterious effects of tobacco use are more visible in the mouth than anywhere else on the body and hence the impact of a BCT in a dental setting may be different than in other settings.

Common tobacco cessation interventions in the dental setting might be classified into three categories, which can be provided alone or in combination with each other. These include pharmacological, educational and behavioral interventions [4]. Pharmacological interventions focus on providing medication, such as Nicotine Replacement Therapies (NRTs), Bupropion and Varenicline. These medications aid in the treatment of tobacco dependence by reducing cravings and relieving withdrawal symptoms [5]. Basic educational interventions involve raising awareness about the harmful effects of tobacco products on general health, whereas behavioral interventions aim to motivate, guide and psychologically assist tobacco users in quitting [2,6].

BCT is a theory-based behavior change method that is used to change one or several psychosocial determinants, such as attitude and self-efficacy [6]. The first BCT taxonomy was developed by Abraham and Michie, who identified 22 BCTs based on an analysis of 221 intervention manuals [7]. Subsequently, a more specific taxonomy was developed to classify the BCTs used in individual behavioral support for smoking cessation: the Behavior Change Techniques Taxonomy for Smoking (BCTTsm) [6]. This taxonomy classifies 44 BCTs based on their function and separates them into four groups: directly addressing motivation (for example, boosting motivation and self-efficacy), maximizing self-regulatory capacity and skills (for example, facilitating relapse prevention and coping), promoting adjuvant activities (for example, advice on stop-smoking medication) and supporting other BCTs (for example, focus on the delivery of the intervention). Previously, the BCTTsm has been applied successfully to classify smoking cessation interventions during pregnancy and text-message-based smoking cessation interventions, and to assess the existing smoking cessation interventions for smokers in the general population [8,9,10].

More evidence on the active ingredients of behavior change interventions for tobacco cessation by oral health professionals (dentists, dental hygienists, dental therapists or dental assistants) is needed. No previous study has reviewed the BCTs used in behavioral interventions for tobacco cessation in dentistry. If we can systematically assess the effects of intervention elements, we can identify what works best, incorporate these elements into future intervention development processes and remove elements that do not add to an intervention impact to lower the user and intermediary burden. Therefore, the aim of this study was to identify the active ingredients (BCTs) used in behavioral interventions for tobacco cessation in dental practices, and explore their impact on intervention effects.

## 2. Methods

### 2.1. Design and Study Selection

This review focused on behavioral interventions for tobacco cessation by oral health professionals. The eligibility criteria for the included studies were as follows: (1) interventions that include effect data, such as randomized or cluster-randomized controlled, quasi-experimental and pretest–posttest evaluations; (2) the intervention group (participants) includes users of any type of tobacco, including smoking and smokeless tobacco; (3) the intervention should be behavioral in nature, but could be combined with educational or pharmacological interventions; (4) the outcome of tobacco cessation at 3, 6 or 12 months after baseline should be reported; and (5) the intervention should be carried out in a dental setting or by oral health professionals.

### 2.2. Strategies for the Literature Search

Search methods covered two main sources: (1) the scientific databases PubMed Central via PubMed, Cochrane, PubMed, Web of Science and PsycINFO and (2) recent reviews/meta-analyses focusing on tobacco cessation in a dental setting. The search strategy within the scientific databases included the use of a combination of keywords and phrases appearing in titles, abstracts or keywords, using the following search algorithm: (a) for describing the participants: tobacco, smoking, cigarette smoking, tobacco smoking, tobacco, smokeless, chewing tobacco, oral tobacco and tobacco users; (b) for describing the intervention: behavio(u)r, (brief) counsel(l)ing, cognitive therapy, dental and patient education; (c) for describing the outcome: tobacco use cessation and tobacco abstinence; (d) for the environment: dentists, hygienists and dental. Interventions were screened first by title, then the abstracts were screened, duplicate manuscripts were removed and eventually the full text was screened. The search only included articles published in English, with no timeframe restriction to reflect the explorative nature of the review and to ensure the inclusion of most evidence-based articles up to the study completion date (7 January 2020). A snowball method was used to retrieve relevant articles from the relevant reviews in addition to the database search. The selection process can be seen in Figure 1, a PRISMA flow diagram [11]. A data extraction form was developed by the authors to gather all the relevant information: intervention name, sample characteristics, intervention contents, comparison group (when applicable) and intervention outcome.

The included interventions were assigned to specific color codes to ease the visual comparison between them and in relation to the evidence strength and the identified BCTs: dark green represented strong evidence, light green represented moderate evidence, orange represented weak evidence, red represented no evidence and gray was used when the OR calculation was not possible (for example, because not all of the relevant information was provided in the manuscript).

### 2.3. Data Analysis

A deductive approach was used by two coders (MI and JM), whereby they identified the BCTs from a predefined set of codes using BCTTsm [6,12]. Following the BCTTsm coding description, the BCT was identified using its code number. Each code was then assigned to one of two overarching themes, direct or indirect, based on its function in terms of tobacco use. BCTs were coded under the direct theme when focusing on aspects related to motivation and regulatory skills of target behavior (tobacco use). When the BCT did not focus on tobacco-use behavior itself, but rather on auxiliary actions related to tobacco use (for example, building a general rapport), it was coded under the indirect theme. Following that, each BCT was matched to one of six sub-codes within each theme. The sub-codes directly addressing motivation and maximizing self-regulatory capacity skills were under the direct theme. Table 1 provides more information and examples of the codes and sub-codes.

Before extracting the data, an inter-coder reliability calculation was performed to measure the degree to which the code generated by the two coders matched after coding the same text independently and without conferring. Two randomly selected interventions were used to assess the inter-coder reliability and the agreement between MI and JM. To determine the overall inter-coder reliability of coding, the overall inter-coder reliability was calculated by dividing the total number of agreements for all codes by the combined total number of agreements and disagreements for all the codes [13,14,15]. The inter-coder reliability for identifying the BCT codes was performed in three rounds. After each round, the inter-coder reliability was measured and followed by a discussion between the coders to reconcile any differences and create a high level of inter-coder consensus. The inter-coder reliability ranged from 73 to 94%, which is considered a high reliability in the literature [13]. After reaching a high level of agreement, the remainder of the included interventions were coded by MI and then reviewed by JM. The intervention effect was expressed as the odds ratio (OR) using the MedCalc’s online calculator [16] and in accordance with the following range: no evidence (OR < = 1), weak evidence (OR = 1.01–2.99), moderate evidence (OR = 3–3.99) and strong evidence (OR > = 4) [17,18]. The choice of OR in the effect measurement was due to the dichotomous nature of the outcome variable.

## 3. Results

Fifteen interventions were included in the review after searching, screening and assessing the eligibility of the articles.

### 3.1. Study Characteristics

Of the fifteen studies included, eleven were randomized controlled trials, one was a randomized uncontrolled trial, one applied a quasi-experimental time series design and two used a pretest–posttest design. Four different countries were identified as follows: country (frequency): USA (11), Sweden (2), UK (1) and India (1) (supplementary Appendix A). The average reported time to deliver the behavioral interventions ranged from three to fifteen minutes. The interventions’ sample sizes ranged from 39 to 3603 participants and included predominantly male samples, with an age range from 13 to 70 years old (supplementary Appendix A). All studies included self-reported tobacco cessation outcome measures, and one study additionally reported carbon monoxide (CO) in exhaled breath. Ten interventions reported significant tobacco use quitting (OR = 2.00–5.25). Two interventions with a pretest–posttest design consisting of a single sample and a binary outcome made it impossible to calculate their effect sizes. Therefore, the data of these two interventions were restricted to identifying and coding the BCTs.

### 3.2. BCTs in Behavioral Interventions for Tobacco Cessation by Oral Health Professionals

Table 2 shows the identified BCTs for the fifteen interventions, which are coded in relation to the frequency of reporting BCTs and intervention effectiveness. Generally, 16 out of 44 smoking-specific BCTs (36.4%) were identified as being applied in one or more interventions. The number of BCTs used in the interventions ranged from 2 to 11 BCTs, with no clear association between the number of BCTs and intervention effectiveness. Across the studies, the most commonly coded BCTs included the following: facilitating goal setting, offering/directing towards appropriate written materials, assessing current readiness and ability to quit, assessing current and past tobacco-use behavior, advising on/facilitating the use of social support, providing feedback on current behavior and advice on stop-tobacco medication (Table 3). Four BCTs were used in effective interventions only, namely, boosting motivation/self-efficacy, identifying reasons for wanting and not wanting to stop using tobacco, facilitating action planning and advising on environmental restructuring (Table 3). The majority (*n* = 12) used oral examinations followed by feedback regarding oral tissue change due to tobacco use.

## 4. Discussion and Conclusions

### 4.1. Discussion

The aim of this review was to identify the active ingredients used in behavioral interventions for tobacco cessation in general, particularly in relation to their effect on tobacco use. Of the 44 BCTTsm, 16 BCTs were identified in tobacco cessation interventions by oral health professionals. More than half (*n* = 10) of these techniques directly focused on the behavior (tobacco use) and aspects relating to motivational and regulatory skills, while the rest of the BCTs promoted auxiliary activities or general interaction with tobacco use. Ten interventions were effective in terms of tobacco use, but there was no clear association between intervention effectiveness and the total number of BCTs applied in the intervention.

An interesting finding that deserves to be pointed out was that the goal-setting technique was included in most of the interventions that showed effectiveness (9/10) in tobacco use cessation and in only two of the ineffective ones (2/3). A previous finding from a systematic review and meta-analysis considered goal setting as a fundamental component for the success of behavior change interventions [34]. Additionally, four BCTs were only used in the effective ones. These four BCTs consisted of boosting self-efficacy, identifying reasons for quitting, action planning and environmental restructuring.

Self-efficacy, or the confidence in one’s ability to quit tobacco use, has received a large amount of attention in many previous studies as a possible predictor of quitting tobacco [35]. Several theories recognized self-efficacy as an important component in their behavior change models, such as the Theory of Planned Behavior and Socio-Cognitive Theory [36,37]. Action planning was also considered an effective strategy in closing the intention–behavior gap. Evidence showed that smoking cessation likelihood increases when action plans are both formulated and enacted [38,39,40]. 

Another noteworthy method that is performed frequently by oral health professionals, in addition to the BCTs, is examinations for oral cancer. Oral cancer examination shows patients the adverse short-term oral health consequences of tobacco use (visible changes in the color of the teeth and the texture of soft tissue), rather than the long-term health consequences, which is considered more effective in deterring tobacco use [4,21,41]. Other variables, such as the degree of nicotine dependence, the number of cigarettes per day, the presence of tobacco-related disease, education level and employment status, might be confounders for the observed effect of BCTs on tobacco cessation [42,43,44]. Moreover, the age and gender of the patient might have also modified the observed effect on tobacco cessation as found in some studies [45,46]. Other studies showed no effect of gender or age on smoking cessation. Furthermore, the rates of smoking cessation in one study were found to be enhanced with age irrespective of gender [43]. Several studies found that education level, employment status and chronic conditions had no observed influence on smoking cessation [45,46,47,48]. However, no clear conclusions can be drawn based on the current findings.

Many interventions were not clear about what the care being received by the comparison group usually entailed. Understanding the active ingredients of both the intervention and usual care groups is important for the effect interpretation [49], as the presence of these active ingredients in the usual care received might also influence the intervention effectiveness. Current guidelines recommend a full reporting of the intervention contents in both the intervention and usual care groups [50]. However, most manuscripts lacked such a detailed account regarding the different circumstances compared.

To the best of our knowledge, this is the first study that identified the BCTs used in tobacco cessation interventions by oral health professionals using BCTTsm. This review made use of reliable and valid evidence-based methods, such as PRISMA and BCTTsm. Furthermore, it focused on identifying the applied BCTs in a dental care setting and not on an analysis of the effectiveness of these BCTs applied in interventions. This review was limited to interventions published in English, meaning it potentially missed interventions published in other languages. However, the evidence showed that excluding studies published in langagues other than English had little effect on the summary of treatment effect estimates [51]. Moreover, a systematic review on empirical studies found no evidence of any systematic bias associated with language restriction in the reviews [52].

### 4.2. Conclusions

The BCTTsm can be used to facilitate the detection of interventions’ active components in a dental setting. Among the behavioral tobacco cessation interventions, certain BCTs were associated with effective tobacco abstinence. Tobacco cessation interventions in a dental setting appear to benefit from using BCTs that increase motivation and teach regulatory skills. Additional interventions that thoroughly report its components and the BCTs used would be valuable to confirm the findings from this review.

### 4.3. Practice Implications

The identified BCTs can provide a source to better inform researchers and oral health professionals about the active ingredients in behavior change interventions for tobacco cessation. Researchers and intervention developers are highly encouraged to provide detailed descriptions of the components in both intervention and comparison groups, possibly in an Open Science Framework to support high-quality reviews on BCT effectiveness in tobacco use cessation. Accurate and complete descriptions of the intervention will be significant for evidence synthesis, as well as for policymakers and clinical practices. Possible research avenues include studies to compare the effectiveness of different BCTs used for tobacco cessation in a dental setting. More tobacco cessation intervention studies are needed in dental settings owing to the uniqueness of this context and its promising result.

## Figures and Tables

**Table 1 ijerph-18-07481-t001:** Codes and sub-codes of the behavior change techniques used in tobacco cessation intervention.

Codes	Sub-Code	Example
Direct BCTs * Focus directly on the behavior	Focus on addressing motivation: Techniques that elicit and support the motivation to quit tobacco	(a) Provide information on the ramifications of smoking and smoking cessation (b) Boost patient confidence on his/her ability to stop tobacco
	Focus on regulatory skills: Techniques that develop skills for planning quitting, identifying barriers and coping with the potential problems	(a) Assist the patient in relapse prevention and coping strategies to avoid the relapses (b) Promote setting a clear action plan to quit that includes, for example, the quitting date
Indirect BCTs Focus indirectly on the behavior	Focus on Adjuvant activities: Techniques that facilitate complimentary support for patient through medication, social network support, websites and telephones.	(a) Explain the benefits, the direction of use as well as the side effects of tobacco cessation medication (b) Promote providing social support for the patient from his/her family and friends
	Focus on delivery of the intervention: Techniques entail general aspects of the interaction and how the intervention was delivered to the patient, including the use of relevant information and considering the patient’s choice of treatment.	(a) Use relevant information from the patient to tailor the behavioral support (b) Emphasize client choice within the evidence-boundaries
	Focus on information gathering: Techniques entail general aspects of the interaction and collecting information about history and experience of tobacco use, tobacco quit attempts and withdrawal symptoms	(a) Assess the pattern of tobacco use frequency, duration and age when started (b) Assess the patient’s current level of readiness to quit
	Focus on general communication: Techniques entail general aspects of the interaction and the provider–patient communication skills	(a) Establish a good professional relationship with the patient (b) Prompt questions eliciting by the patient

***** BCTs: behavior change techniques. This table followed the description of the taxonomy of Behavior Change Techniques used in individual behavioral support for smoking cessation by Michie and colleagues [6].

**Table 2 ijerph-18-07481-t002:** Behavior change techniques for tobacco cessation interventions in dental practice and its relation to intervention effect.

Behavior Change Technique	Code	Studies (*n* = 15)														
		Gordon 2005 [19]	Severson 2009 [20]	Virtanen 2015 [21]	Gordon 2010 [22]	Andrews 1999 [23]	Cohen 1989 [24]	Stevens 1995 [25]	Walsh 2003 [26]	Severson 1998 [27]	Nohlert 2009 [28]	Binnie 2007 [29]	Greene 1994 [30]	Gansky 2005 [31]	Gonseth 2010 [32]	Secker-Walker 1988 [33]
**1.Direct: Focus on behavior**																
**Specific focus on behavior (B) and motivation (M)**																
Provide information on consequences of tobacco use and tobacco cessation	BM1															
Boost motivation and self-efficacy	BM2															
Provide feedback on current behavior	BM3															
Identify reasons for wanting and not wanting to stop using tobacco	BM9															
Measure CO *	BM11															
**Specific focus on behavior (B) and regulatory skills (S)**																
Facilitate relapse prevention and coping	BS2															
Facilitate action planning	BS3															
Facilitate goal setting	BS4															
Advise on changing routine	BS7															
Advise on environmental restructuring	BS8															
**2.Indirect: Promote adjuvant activities (A)**																
Advise on stop-tobacco medication	A1															
Advise on/facilitate use of social support	A2															
**3.Indirect: General aspects of interaction (R)**																
**General aspects of interaction (R) focusing on delivery of the intervention (D)**																
Tailor interactions appropriately	RD1															
**General aspects of interaction (R) focusing on information gathering (I)**																
Assess current and past tobacco-use behavior	RI1															
Assess current readiness and ability to quit	RI2															
**General aspects of interaction (R) focusing on general communication (C)**																
Offer/direct towards appropriate written materials	RC5															
CO * = Carbon monoxide measured in exhaled breath.																
	*Strong Evidence (OR >= 4)*				*Moderate Evidence* *(OR = 3–3.99)*			*Weak Evidence (OR = 1.01–2.9)*			*No Evidence* *(OR <= 1)*				*Effect calculation* *is not possible*	

**Table 3 ijerph-18-07481-t003:** Coding table of frequently identified behavior change techniques.

List of Codes	Frequency	Example Quotes from Included Studies
Assess current and past tobacco use behavior	11	“asking all patients about their tobacco use” [19]
Advise on stop-tobacco medication	8	“the intervention participants received free nicotine replacement therapy (NRT) in the form of patches or gum, as part of their treatment plan” [19]
Facilitate goal setting	12	“setting a quit date, developing a plan, and training in action and thinking skills to get ready to quit and to prevent relapse” [21] “mutually agree on a quit date”[27]
Offer/direct towards appropriate written materials	12	“at the end of this brief counseling session the patient was given a brief self-help booklet” [31]
Assess current readiness and ability to quit	12	“assessing readiness to quit via brief verbal questions” [26]
Advise on/facilitate use of social support	10	“a second sheet was available for patient’s partner which outlined ways to support quit attempt and discuss strategies for quitting together” [28]
Provide feedback on current behavior	9	“your use of smokeless tobacco is probably related to this precancerous lesion here in your mouth” [22]

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
