# Peer review of "Identifying Behavior Change Techniques Used in Tobacco Cessation Interventions by Oral Health Professionals and Their Relation to Intervention Effects—A Review of the Scientific Literature"

_ijerph, 2021, doi:10.3390/ijerph18147481_

Round 1
Reviewer 1 Report
I have read the article "Identifying Behavior Change Techniques Used in Tobacco Cessation Interventions by Oral Health Professionals and their Relation to Intervention Effect; A Review of the Scientific Literature." very carefully. The authors raise interesting issues in a concise and accessible way. The article is an interesting and new contribution to the field of Behavior Change Techniques Used in Tobacco Cessation Interventions. I only propose that the authors consider the use of more precise wording in lines 64-68. On the one hand, the authors state that there are no current studies identifying the BCTs used in behavioral interventions for tobacco cessation in dentistry. On the other hand, they point to the research objective to identify the active ingredients (BCTs) used in behavioral interventions for tobacco cessation in dental practices, and explore their impact on intervention effects. In my opinion, these are mutually exclusive statements. Since there are no currently available studies, it is difficult to talk about such a formulated aim of the study.
Author Response
Point 1: Reviewer 1: I have read the article "Identifying Behavior Change Techniques Used in Tobacco Cessation Interventions by Oral Health Professionals and their Relation to Intervention Effect; A Review of the Scientific Literature." very carefully. The authors raise interesting issues in a concise and accessible way. The article is an interesting and new contribution to the field of Behavior Change Techniques Used in Tobacco Cessation Interventions. I only propose that the authors consider the use of more precise wording in lines 64-68. On the one hand, the authors state that there are no current studies identifying the BCTs used in behavioral interventions for tobacco cessation in dentistry. On the other hand, they point to the research objective to identify the active ingredients (BCTs) used in behavioral interventions for tobacco cessation in dental practices, and explore their impact on intervention effects. In my opinion, these are mutually exclusive statements. Since there are no currently available studies, it is difficult to talk about such a formulated aim of the study.
Response 1: Thank you for your compliment and we appreciate your comments. We incorporated your suggestion by rephrasing the statement: “No previous study has reviewed the BCTs used in behavioral interventions for tobacco cessation in dentistry.” [Lines 64-65]

Reviewer 2 Report
Can you discuss other variables that may influence the intervention, for example, dependence on nicotine, sex, age, cigarette per day, etc.?
Author Response
Response to Reviewer 2 Comments
Point 1: Reviewer 2: Can you discuss other variables that may influence the intervention, for example, dependence on nicotine, sex, age, cigarette per day, etc.?
Response 1: Thank you for your comment. We have incorporated your suggestion in the discussion session: “Other variables as the degree of nicotine dependence, the number of cigarettes per day, presence of tobacco-related disease, education level and employment status might be confounders for the observed effect of BCTs on tobacco cessation [43-45]. Moreover, age and gender of the patient might have also modified the observed effect on tobacco cessation as found in some studies [46, 47]. Other studies showed no effect of gender or age on smoking cessation. Furthermore, the rates of smoking cessation in one study were found to be enhanced with age irrespective of gender [44]. Several studies found that education level, employment status and chronic conditions had no observed influence on the smoking cessation. However, no clear conclusions can be drawn based on the current findings [46-49].” [Lines 206-215].

Reviewer 3 Report
The paper deals with a specific topic, I don't know how relevant it could be. However, it might be of interest for some health professionals. The introduction and method section could be better contextualized. I appreciated the discussion and conclusion section.
Line 39-40: “Common tobacco cessation interventions in the dental setting might be classified into three categories, which could be provided alone or in combination with each other. These include pharmacological, educational and behavioral interventions.” I suggest to provide a reference for this
Line 44-48: “Basic educational interventions include raising awareness about the harmful effect of tobacco products on general health, whereas behavioral interventions aim to motivate, guide and psychologically assist tobacco users in quitting [2, 5]. BCT is a theory-based behavior change method that is used to change one or several psychosocial determinants, such as awareness and self-efficacy [5]” The authors use “awareness” to describe both educational interventions and behavioral interventions; I suggest to use two different terms, in order to clarify the differences between the two types of interventions. Below, in the BCTTsm description, awareness is not listed.
Line 49-50: “The first BCT taxonomy was developed by Abraham and Michie, who identified 22 BCTs based an analysis of 221 intervention manuals [6]” I think a proofread of the English is needed here
Line 62-63: “More evidence on the active ingredients of behavior change interventions for tobacco cessation by oral health professionals (dentists, dental hygienists, dental therapists or dental assistants) is needed.” Why? In order to give strength to the study aim, the authors should better describe why it is necessary to collect more evidence in this regard.
Line 114-117: “The sub-codes: directly addressing motivation and maximizing self-regulatory capacity skills 115 were under the direct theme. While the indirect theme encompassed promoting adjuvant activities, focusing on intervention delivery, focusing on gathering information and focusing on general communication.” I suggest to explain what each sub-code refers to (for instance, what does “focusing on general communication” refers to?), and the way it is related to tobacco cessation.
Line 189-191: “Additionally, four BCTs – boosting self-efficacy, identifying reasons for quitting, action planning and environmental restructuring – only used in the effective ones.” Maybe this statement should be rephrased.
Author Response
Response to Reviewer 3 Comments
Point 1:
Line 39-40: “Common tobacco cessation interventions in the dental setting might be classified into three categories, which could be provided alone or in combination with each other. These include pharmacological, educational and behavioral interventions.” I suggest to provide a reference for this.
Response 1: Thank you for your suggestion. We added the reference: [4] [Line 41]
Point 2: Line 44-48: “Basic educational interventions include raising awareness about the harmful effect of tobacco products on general health, whereas behavioral interventions aim to motivate, guide and psychologically assist tobacco users in quitting [2, 5]. BCT is a theory- based behavior change method that is used to change one or several psychosocial determinants, such as awareness and self-efficacy [5]” The authors use “awareness” to describe both educational interventions and behavioral interventions; I suggest to use two different terms, in order to clarify the differences between the two types of interventions. Below, in the BCTTsm description, awareness is not listed.
Response 2: We appreciate your comment. Your suggestion was incorporated to the introduction: “BCT is a theory-based behavior change method that is used to change one or several psychosocial determinants, such as attitude and self-efficacy [6]” [Lines 48- 49]
Point 3: Line 49-50: “The first BCT taxonomy was developed by Abraham and Michie, who identified 22 BCTs based an analysis of 221 intervention manuals [6]” I think a proofread of the English is needed here
Response 3: Thank you for your comment. The paragraph was rephrased by native English speaker as: “The first BCT taxonomy was developed by Abraham and Michie, who identified 22 grouped BCTs based on an analysis of 221 intervention manuals [7]” [Lines 49-51]
Point 4: Line 62-63: “More evidence on the active ingredients of behavior change interventions for tobacco cessation by oral health professionals (dentists, dental hygienists, dental therapists or dental assistants) is needed.” Why? In order to give strength to the study aim, the authors should better describe why it is necessary to collect more evidence in this regard.
Response 4: Thank you for your comment. We have incorporated an explanation as suggested: “if we can systematically assess the effect of intervention elements, we can identify what works (best), incorporate these elements in future intervention development processes and remove those elements that do not add to an intervention impact to lower user and intermediary burden.” [Lines 65-68]
Point 5: Line 114-117: “The sub-codes: directly addressing motivation and maximizing self- regulatory capacity skills 115 were under the direct theme. While the indirect theme encompassed promoting adjuvant activities, focusing on intervention delivery, focusing on gathering information and focusing on general communication.” I suggest to explain what each sub-code refers to (for instance, what does “focusing on general communication” refers to?), and the way it is related to tobacco cessation.
Response 5: Thank you for your suggestion. We have included definition for each sub-code in one table Table 1.
Table 1. Codes and Sub-codes of the Behavior Change Techniques used in Tobacco Cessation Intervention.
| Codes | Sub-code | Example |
|
Direct BCTs* Focus directly on the behavior
|
Focus on addressing motivation: Techniques that elicit and support the motivation to quit tobacco
|
a) Provide information on the ramifications of smoking and smoking cessation. b) Boost patient confidence on his/her ability to stop tobacco
|
|
Focus on regulatory skills: Techniques that develop skills for planning quitting, identifying barriers, and coping with the potential problems
|
a) Assist the patient in relapse prevention and coping strategies to avoid the relapses b) Promote setting a clear action plan to quit that include for example the quitting date
|
|
|
Indirect BCTs Focus indirectly on the behavior |
Focus on Adjuvant activities:Techniques that facilitate complimentary support for patient through medication, social network support, websites, and telephones. | a) Explain the benefits, the direction of use as well as the side effects of tobacco cessation medication. b) Promote providing social support for the patient from his/her family and friends |
| Focus on delivery of the intervention: Techniques that entail general aspect of the interaction and how the intervention was delivered to the patient including the use of relevant information and considering patient choice of treatment. | a) Use relevant information from the patient to tailor the behavioral support. b) Emphasise client choice within the evidence- boundaries | |
| Focus on information gathering: Techniques that entail general aspect of the interaction and collecting information about history and experience of tobacco use, tobacco quit attempts and withdrawal symptoms. | a) Assess the pattern of tobacco use frequency, duration and age when started. b) Assess patient current level of readiness to quit | |
| Focus on general communication: Techniques that entail general aspect of the interaction and the provider-patient communication skills | a) Establish good professional relationship with the patient. b) Prompt questions eliciting by the patient |
Point 6: Line 189-191: “Additionally, four BCTs – boosting self-efficacy, identifying reasons for quitting, action planning and environmental restructuring – only used in the effective ones.” Maybe this statement should be rephrased.
Response 6: Thank you for your comment. We have rephrased the sentence to: “Additionally, four BCTs only used in the effective ones. These four BCTs consisted of boosting self-efficacy, identifying reasons for quitting, action planning and environmental restructuring.” [Lines 192- 194]

Round 2
Reviewer 3 Report
The quality of the work ameliorated after the changes. Thank you